# A Trajectory Generation Algorithm for a Re-Entry Gliding Vehicle Based on Convex Optimization in the Flight Range Domain and Distributed Grid Points Adjustment

**Mingjie Li [1], Chijun Zhou [2,*], Lei Shao [2], Humin Lei [2] and Changxin Luo [2]**

[1]   Graduate College, Air Force Engineering University, Xi'an 710051, China
[2]   Air Defense and Missile Defense College, Air Force Engineering University, Xi'an 710051, China
*   Correspondence: zhouchijun666@126.com

**Featured Application: The method proposed can be used for accurate trajectory generation of a re-entry glide vehicle in the flight range domain with distributed grid points. It also provides ideas for trajectory generation based on the guidance mechanism.**

**Abstract:** Optimal trajectory generation for the guidance of re-entry glide vehicles is of great significance. To realize a faster generation speed and consistency with the guidance mechanism, an improved convex optimization trajectory generation algorithm based on the flight range domain for the re-entry glide vehicles is proposed in this paper. Firstly, according to the definition of the range-to-go, the projected range-to-go of the re-entry glide vehicle is presented when the dynamic model is converted to the flight range domain. Then, the attack angle and bank angle are expanded to the state variables and the change rate, which is designed as a new control variable. When the dynamic models and constraints are convexificated and discretized, the vehicle trajectory discrete convex model in the flight range domain is proposed. In order to further improve the generation speed and accuracy, an initial trajectory generation method that is close to the guidance requirements is proposed by the landing points of different control laws. In addition, by analyzing the nonlinear illegal degree of grid points, the probability density of grid points and the adjustment strategy of grid points are proposed. Finally, the ablation experiment shows that the initial trajectory generation and distributed grid points method works. With different target points, different no-fly zones, different initial states, and the Monte Carlo experiment, our method can effectively and robustly complete the generation. The lateral and longitudinal generation error is less than 1 km. And compared with the Gaussian pseudo-spectral method, our method obtained comparable accuracy and faster speed.

**Keywords:** trajectory generation for re-entry glide vehicle; sequence convex optimization; dynamic model in the fight range domain; distributed grid points adjustment





## 1. Introduction

The re-entry glide vehicle [1] with a lift body structure has extremely fast re-entry and glide speeds and can make a jump maneuver of tens of kilometers in the longitudinal direction with hundreds of kilometers of maneuvering range in the lateral direction, which is difficult to predict and intercept [2], and thus has high research value.

Research on the trajectory generation mechanism of re-entry glide vehicles is beneficial to generating more effective re-entry trajectories and improving guidance accuracy. In general, the trajectory generation methods of re-entry glide vehicles are mainly divided into predictor–corrector guidance and nominal trajectory guidance. At present, the numerical predictor–corrector method is adopted within the mainstream predictor–corrector guidance methods, and the process is mainly divided into longitudinal and lateral guidance. Longitudinal guidance is primarily realized by setting the attack angle profile and using the secant method [3], the landing points method [4], or the flight range prediction network [5],

to calculate the value of the bank angle. In the lateral direction, in order to consider the guidance and the avoidance of the no-fly zones, the avoidance logic [6] is usually combined with the heading deviation angle corridor [7] and cross-range corridor [8] when the adaptive cross-range corridor [5] integrating the two functions is also used to realize guidance. In general, although the trajectory generation speed of the predictor-corrector algorithm is fast, its guidance accuracy can be further improved.

At the same time, the nominal trajectory generation of a re-entry glide vehicle is a complex nonlinear problem. The primary method is transforming the problem of re-entry trajectory generation into the optimization problem of discrete points through the sequential convex optimization method [9]. In recent years, several types of research have been performed. Liu [10] et al. and Wang [11] et al. solved the multiple constraint re-entry optimization problem based on the sequential quadratic programming (SQP) method through linearization and variable relaxation. Wang [12] et al. treated the online trajectory generation problem of re-entry glider aircraft as a second-order cone programming problem and designed the corresponding optimal feedback tracking guidance design for effective tracking. Hong [13] et al. realized continuous guidance of re-entry vehicles based on the convex optimization algorithm and continuous closed-loop control. Liu [14,15] et al. introduced a nonlinear relaxation method, which can reduce the relaxation degree, and the effectiveness of the method is verified. Wang [16] et al. used the convexity method to provide a high-quality initial trajectory guess for the pseudo-spectral method. All the above methods realized the convexification of the re-entry dynamic model and constraints of the vehicle in the time domain and obtained the generated re-entry trajectory. Sandberg [17] et al. compared the optimal trajectories using Pontryagin's method and the slew trajectories using sinusoidal functions, and approximately 1.5% lower control effort was obtained. Then, based on Sandberg's approach, Raigoza [18] established an autonomous trajectory maneuver to de-orbit spacecraft back to Earth with a distributed waypoint for autonomous collision avoidance. However, according to the guidance mechanism, the planned spatial state of grid points is not directly reflected in the time domain, more states can be considered to describe the dynamic model.

In order to improve the convergence and accuracy of trajectory generation based on convex programming, many improved methods have been proposed. Wang [19] and Zhou [20,21] introduced new variables to adjust the trust region in the aircraft planning process adaptively, and this can realize faster convergence. Saglino [22] et al. converted the guidance problem of the re-entry glide vehicle to the energy domain, reduced the non-convexity of the equation, and solved it with the pseudo-spectral method. Liu [14] et al. realized the conversion of the guidance problem of the vehicle to the altitude domain and also realized the practical re-entry guidance. From the above methods, it can be seen that changing the described domain of dynamic equations may realize effective convexity and convergence. At the same time, the operation of the variable trust region is also worded to increase the convergence accuracy of the generated trajectory.

For the grid point adjustment method of re-entry trajectory generation, the Gaussian and Radau pseudo-spectral method are proven to be effective [23]. The hp pseudo-spectral method is used to select the root of the orthogonal polynomial as the grid points [24], which can be dynamically adjusted with iteration. Li [25] et al. proposed a multi-segment grid point modified Radau pseudo-spectral method to realize the generation and fast convergence of convex optimized trajectory. However, the pseudo-spectral method is better for linear problems and convex constraints and is not fully applicable to sequential planning problems. In addition to the pseudo-spectral method, some adaptive grid points adjustment methods are also proposed. Zhao [26] et al. proposed an adaptive grid point adjustment method, which solved the problem of Mars re-entry trajectory optimization by analyzing iteration errors to realize insertion and deletion points. Zhou [27] put forward the concept of nonlinear illegal degree and used the error threshold to be the insertion and deletion rules of grid points, thus realizing the dynamic adjustment of grid points. The concept of nonlinear illegal degree proposed by the above methods has an excellent guiding

significance for grid point adjustment, but the grid point adjustment method is fixed, and the increase or decrease may be significantly affected by the convergence process.

To solve the problem that the description of re-entry trajectory generation in the time domain is not direct enough, the initial trajectory of iteration affects the convergence speed, and the grid point adjustment mode is fixed, a new convex optimized trajectory generation with the flight range domain and distributed grid points adjustment are proposed in this paper. Firstly, based on the definition of range-to-go, the concept of projected range-to-go is proposed, by which the dynamic model of the vehicle is transformed to the flight range domain for representation. At the same time, the states of the vehicle are expanded, and the dynamic model and constraints of the glider are convex and discrete. Then, the re-entry trajectory generation problem is transformed into a sequential programming problem. In order to improve the solving speed, the initial trajectory that basically meets the guidance requirements is obtained through transformation and interpolation according to the landing points of the vehicle under different constant control laws. In order to further improve the trajectory generation accuracy and speed up the trajectory convergence, a grid points probability density function is proposed according to the nonlinear illegal degree of the iterative trajectory, which is used to change the rules of generating new grid points and adjust the location of grid points. Through ablation experiments, the effectiveness of the improved part is proved. Additionally, by changing the initial and terminal positions and conducting Monte Carlo experiments, the robustness of our method is verified. The generation errors of our method are less than 1 km. Finally, through comparison with other mainstream methods, we prove the superiority of our method in search speed and accuracy.

The innovation of this paper mainly includes the following three parts:

1.  According to the concept of range-to-go in the guidance of re-entry glide vehicles, the projected range-to-go is proposed. By the definition of projected range-to-go, the dynamic model of the vehicle is transformed from the time domain to the flight range domain. Then the dynamic model and constraints are convexification and discretization, and the final sequential convex optimization expression is proposed;
2.  According to the landing points under the different constant control laws, the initial trajectory generation problem of the vehicle in any spatial state is transformed into a similar initial state trajectory generation problem by rotation transformation. And the initial trajectory that can basically realize practical guidance and meet the dynamic model is obtained by interpolation;
3.  According to the concept of the nonlinear illegal degree of iteration trajectory and the distribution of the nonlinear illegal degree, the grid points probability density function is proposed, and the grid points adjustment law is proposed by the probability density function, which realizes efficient and fast grid points adjustment.

The section of this paper is arranged as follows: Section 1 describes the research status of convex trajectory generation of re-entry glide vehicles and the innovation of this paper; in the Section 2, the dynamic model and constraints of the trajectory generation in the flight range domain are described; in the Section 3, the convexity and discretization process of the trajectory programming in the flight range domain are described; in the Section 4, the initial trajectory generation method and grid point adjustment strategy are proposed; in the Section 5, operation of the simulation is used to verify the research content; the Section 6 summarizes the conclusion of this paper.

## 2. Dynamic Models and Constraints of Re-Entry Glide Vehicle

### 2.1. Dimensionless Dynamic Model in the Flight Range Domain

In order to facilitate the trajectory generation of the vehicle, the dynamic model of the vehicle is usually dimensionless in the time domain. The dimensionless dynamic equation is

$$\begin{cases} \dot{r} = v \sin\theta \\ \dot{\phi} = v \cos\theta \sin\psi / r \cos\varphi \\ \dot{\varphi} = v \cos\theta \cos\psi / r \\ \dot{v} = -D - \sin\theta / r^2 \\ \dot{\theta} = L\cos\beta / v + v\cos\theta / r + \cos\theta / (vr^2) + C_\theta + \widetilde{C}_\theta \\ \dot{\psi} = L\sin\beta / (v\cos\theta) + v\cos\theta\sin\psi\tan\varphi / r + C_\psi + \widetilde{C}_\psi \end{cases} \tag{1}$$

where, $r$ represents the dimensionless geocentric distance of the vehicle, $\phi$, $\varphi$ represent the longitude and latitude coordinates, $v$ represents the dimensionless speed, $\theta$ represents the flight path angle, $\psi$ represents the flight heading angle of the target, $L$ and $D$ represent the dimensionless lift and drag of the vehicle, $C_\theta$, $\widetilde{C}_\theta$ and $C_\psi$, $\widetilde{C}_\psi$ represent the coriolis acceleration item and the involved acceleration item caused by the earth's rotation.

$$\begin{cases} L = \left(\rho(v \cdot v_c)^2 SC_L\right) / 2mg_0 \\ D = \left(\rho(v \cdot v_c)^2 SC_D\right) / 2mg_0 \end{cases} \tag{2}$$

where, $C_L$ represents the lift coefficient, $C_D$ represents the drag coefficient, $S$ represents the reference area of the vehicle, $m$ represents the mass of the vehicle, and $g_0$ represents the gravity acceleration at zero altitude, $v_c = \sqrt{g_0 R_e}$ represents the dimensional velocity where $R_e$ is the radius of earth, $\rho$ is the atmosphere density. The dimensionless operation of other quantities can be referred to in the literature [28].

Define the initial range-to-go of the re-entry glide vehicle as $S_{f0}$.

$$S_{f0} = \arccos\left(\cos\varphi_f \cos\varphi_0 \cos(\phi_f - \phi_0) + \sin\varphi_f \sin\varphi_0\right) \tag{3}$$

where, $\phi_0$, $\varphi_0$ represent the longitude and latitude coordinates of the initial point, and $\phi_f$, $\varphi_f$ represent the longitude and latitude coordinates of the target point.

Then the speed of the vehicle between the initial point and the target point is $v_s = v\cos\theta\cos(\psi - \psi_p)$, as shown in Figure 1. When the initial point and target point are given, the problem of effective trajectory generation for the vehicle can be regarded as a problem of decreasing the projected range-to-go of the vehicle between the initial point with the target point. If the projected range-to-go is $s$, then

$$ds/dt = -v\cos\theta\cos(\psi - \psi_p)/r \tag{4}$$

where $\psi_p$ represents the flight heading angle of the target point relative to the initial point, $\psi_p = \arcsin\left(\sin(\phi_f - \phi_0)\cos\varphi_f / \sin S_f\right)$.

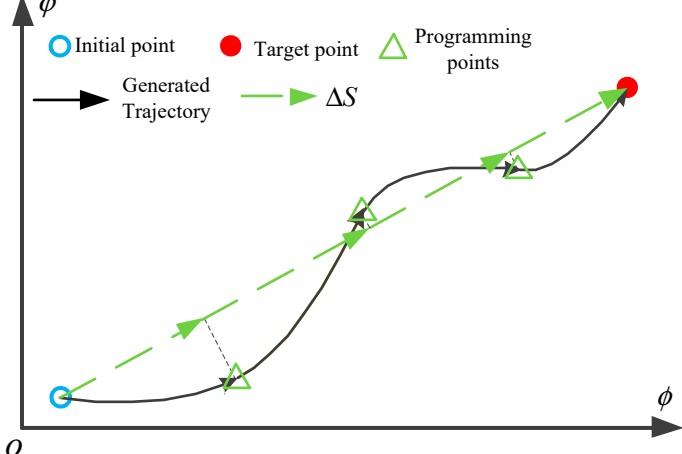

**Figure 1.** Generated trajectory in the flight range domain.

Operating Equation (1) with Equation (4), the dynamic model of the vehicle in the flight range domain can be expressed as

$$
\begin{cases}
dr/ds = -r\tan\theta/\cos(\psi - \psi_p) \\
d\phi/ds = -\sin\psi/\left(\cos\varphi\cos(\psi - \psi_p)\right) \\
d\varphi/ds = -\cos\psi/\cos(\psi - \psi_p) \\
dv/ds = Dr/v\cos\theta\cos(\psi - \psi_p) + \tan\theta/vr\cos(\psi - \psi_p) \\
d\theta/ds = -L\cos\beta r/\left(v^2\cos\theta\cos(\psi - \psi_p)\right) - 1/\cos(\psi - \psi_p) + 1/\left(v^2 r\cos(\psi - \psi_p)\right) + \left(C_\theta + \widetilde{C}_\theta\right) \\
d\psi/ds = -L\sin\beta r/\left(v^2\cos^2\theta\cos(\psi - \psi_p)\right) - \sin\psi\tan\varphi/\left(\cos(\psi - \psi_p)\right) + \left(C_\psi + \widetilde{C}_\psi\right)
\end{cases}
\tag{5}
$$

The corresponding acceleration term can be converted into

$$
\begin{cases}
C_\theta = -2r\omega_e\sin\psi\cos\varphi/\left(v\cos\theta\cos(\psi - \psi_p)\right) \\
C_\theta' = -\omega_e^2 r^2(\cos\varphi\sin\varphi\cos\psi\sin\theta + \cos^2\phi\cos\theta)/\left(v^2\cos\theta\cos(\psi - \psi_p)\right) \\
C_\psi = -2r\omega_e(\sin\varphi - \cos\varphi\tan\theta\cos\psi)/\left(v\cos\theta\cos(\psi - \psi_p)\right) \\
C_\psi' = -\omega_e^2 r^2\sin\varphi\cos\varphi\sin\psi/v^2\cos^2\theta\cos(\psi - \psi_p)
\end{cases}
\tag{6}
$$

where $\omega_e$ represents the earth rotation constant.

### 2.2. Constraints Settings

There are process constraints, no-fly zone constraints, and terminal constraints in the flight of the vehicle. Where the terminal constraint is expressed as

$$
\begin{cases}
r(s = 0) = r_f \\
v(s = 0) = v_f \\
\phi(s = 0) = \phi_f \\
\varphi(s = 0) = \varphi_f
\end{cases}
\tag{7}
$$

where, $r_f$, $v_f$, $\phi_f$, $\varphi_f$ represent the terminal altitude, terminal speed and longitude and latitude, respectively.

The process constraint includes the control variables constraint, heat flux constraint $\dot{Q}_{max}$, dynamic pressure constraint $q_{max}$, and overload constraint $n_{max}$, of the vehicle. The control variables generally include the attack angle $\alpha$ and bank angle $\beta$ constraints

$$
\begin{cases}
\alpha_{min} \le \alpha \le \alpha_{max} \\
\beta_{min} \le \beta \le \beta_{max}
\end{cases}
\tag{8}
$$

where, $\alpha_{min}, \alpha_{max}, \beta_{min}, \beta_{max}$, represent the minimum, the maximum value of attack angle, the minimum and the maximum value of the bank angle, respectively.

The heat flow density, dynamic pressure constraint and overload calculation formulas are shown as

$$
\begin{cases}
\dot{Q} = C\rho^{0.5}(v \cdot v_c)^{3.15} \le \dot{Q}_{max} \\
q = 0.5\rho(v \cdot v_c)^2 \le q_{max} \\
n = \sqrt{L^2 + D^2}r^2 \le n_{max}
\end{cases}
\tag{9}
$$

where, $\dot{Q}_{max}$ represents the maximum value of heat flow density, $q_{max}$ represents the maximum value of dynamic pressure, $n_{max}$ represents the maximum value of overload, $C$ represents the aerodynamic heat coefficient.

In this paper, the no-fly zone is set as a circular area, and the re-entry glide vehicle shall not pass through the no-fly zone. It can be expressed as

$$
(\phi - \phi_{bi})^2 + (\varphi - \phi_{bi})^2 - r_{bi}^2 \ge 0
\tag{10}
$$

where, $\phi_b$ and $\varphi_b$ represent the central longitude and latitude of the center of no-fly zone, $r_b$ represent the longitude and latitude radius of the no-fly zone, and $i = 1, \ldots, n$ represent the number of no-fly zones.

### 2.3. Description of Re-Entry Generation Problem in the Flight Range Domain

The purpose of trajectory generation is to realize terminal constraints under several constraints. Thus, the optimization objective is

$$J_0 = \kappa_1 \left( (\phi_f^s - \phi_f)^2 + (\varphi_f^s - \varphi_f)^2 \right) + \kappa_2 (r_f^s - r_f)^2 \tag{11}$$

where, $\phi_f^s, \varphi_f^s, r_f^s$ represent longitude and latitude and geocentric distance when the projected range-to-go of the vehicle is 0. $\kappa_1, \kappa_2$ are optimization coefficients.

The optimization problem of trajectory generation in the flight range domain can be expressed as

$$\begin{aligned} P_0 : \min \quad & J_0 = \kappa_1 \left( (\phi_f^s - \phi_f)^2 + (\varphi_f^s - \varphi_f)^2 \right) + \kappa_2 (r_f^s - r_f)^2 \\ \text{subject to}: \quad & (5), (8), (9), (10), v(s=0) = v_f \end{aligned} \tag{12}$$

## 3. Convexification and Discretization of the Re-Entry Trajectory Generation Problem

The optimization problem in the Section 2 is a nonlinear and non-convex problem, which is difficult to solve directly. Therefore, the problem needs to be convex.

### 3.1. State, Control Variable Settings and Model Convexity

According to the equation dynamic model (5), the control variables are the attack angle and bank angle, while the lift and drag coefficients of the CAV [29], like the vehicle simulated in this paper generally meet the following equations:

$$\begin{cases} C_L = c_{l2}\alpha + c_{l1} \\ C_D = c_{d3}\alpha^2 + c_{d2}\alpha + c_{d1} \end{cases} \tag{13}$$

where, $c_{l1}, c_{l2}$ represent the lift parameters, and $c_{d1}, c_{d2}, c_{d3}$ represent the drag parameters.

Set the state variable $X = [r, \phi, \varphi, v, \theta, \psi, \alpha, \beta]$ and control variable $U = [\dot{\alpha}, \dot{\beta}]$ of the vehicle. With the new control quantity, the dynamic model (5) is converted into

$$\begin{cases} dr/ds = -r \tan\theta / \cos(\psi - \psi_p) \\ d\phi/ds = -\sin\psi / (\cos\varphi \cos(\psi - \psi_p)) \\ d\varphi/ds = -\cos\psi / \cos(\psi - \psi_p) \\ dv/ds = C_D rqs / mg_0 v \cos\theta \cos(\psi - \psi_p) + \tan\theta / (vr \cos(\psi - \psi_p)) \\ d\theta/ds = -C_L \cos\beta rqs / (mg_0 v^2 \cos\theta \cos(\psi - \psi_p)) \\ \qquad\qquad -1/\cos(\psi - \psi_p) + 1/(v^2 r \cos(\psi - \psi_p)) + \left( C_\theta + \widetilde{C}_\theta \right) \\ d\psi/ds = -C_L \sin\beta rqs / (mg_0 v^2 \cos\theta \cos(\psi - \psi_p)) \\ \qquad\qquad -\sin\psi \tan\varphi / (\cos(\psi - \psi_p)) + \left( C_\psi + \widetilde{C}_\psi \right) \\ d\alpha/ds = \dot{\alpha} \\ d\beta/ds = \dot{\beta} \end{cases} \tag{14}$$

Equation (14) can be expressed as

$$\dot{X} = g(X, S) + BU(S) + h(X, S) = F(X, U, S) \tag{15}$$

$$g(X,S) = \begin{bmatrix} -r\tan\theta/\cos(\psi-\psi_p) \\ -\sin\psi/(\cos\varphi\cos(\psi-\psi_p)) \\ -\cos\psi/\cos(\psi-\psi_p) \\ C_D rqs/mg_0 v\cos\theta\cos(\psi-\psi_p) + \tan\theta/(vr\cos(\psi-\psi_p)) \\ -C_L\cos\beta rqs/mg_0 v^2\cos\theta\cos(\psi-\psi_p) - \frac{r}{v^2\cos(\psi-\psi_p)}(\frac{v^2}{r}-\frac{1}{r^2}) \\ -C_L\sin\beta rqs/mg_0 v^2\cos^2\theta\cos(\psi-\psi_p) - \sin\psi\tan\varphi/\cos(\psi-\psi_p) \\ 0 \\ 0 \end{bmatrix} \tag{16}$$

$$B = \begin{bmatrix} 0 & 0 & 0 & 0 & 0 & 0 & 1 & 0 \\ 0 & 0 & 0 & 0 & 0 & 0 & 0 & 1 \end{bmatrix}^T \tag{17}$$

$$h(X,S) = \begin{bmatrix} 0\,0\,0\,0\,C_\theta + C_\theta'\ C_\psi + C_\psi'\,0\,0 \end{bmatrix}^T \tag{18}$$

Since $g(X,S)$ is a nonlinear function of $X$, further linearization of the dynamic equation is required. The first order Taylor expansion is operated on the last iterative trajectory, which can be expressed as

$$\dot{X} = \left.\frac{\partial g(X)}{\partial X}\right|_{x=x^{(k)}} X + BU + g\left(X^{(k)}\right) - \left.\frac{\partial g(X)}{\partial X}\right|_{X=X^{(k)}} X^{(k)} + h(X^{(k)}) \tag{19}$$

Let $A = \left.\frac{\partial g(x)}{\partial x}\right|_{x=x^{(k)}}$, $C = g\left(X^{(k)},S\right) - AX^{(k)} + h(X^{(k)},S)$, then

$$\dot{X} = AX + BU + C \tag{20}$$

$$A = \begin{bmatrix} a_{11} & 0 & 0 & 0 & a_{15} & a_{16} & 0 & 0 \\ 0 & 0 & a_{23} & 0 & 0 & a_{26} & 0 & 0 \\ 0 & 0 & 0 & 0 & 0 & a_{36} & 0 & 0 \\ a_{41} & 0 & 0 & a_{44} & a_{45} & a_{46} & a_{47} & 0 \\ a_{51} & 0 & 0 & a_{54} & a_{55} & a_{56} & a_{57} & a_{58} \\ a_{61} & 0 & a_{63} & 0 & a_{65} & a_{66} & a_{67} & a_{68} \\ 0 & 0 & 0 & 0 & 0 & 0 & 0 & 0 \\ 0 & 0 & 0 & 0 & 0 & 0 & 0 & 0 \end{bmatrix} \tag{21}$$

$$\begin{cases} a_{11} = -\tan\theta^{(k)}/\cos\left(\psi^{(k)} - \psi_p\right) \\ a_{41} = D^{(k)}\left(1 - R_e r^{(k)}/H_0\right)/\left(v^{(k)}\cos\theta^{(k)}\cos\left(\psi^{(k)} - \psi_p\right)\right) \\ \qquad - \tan\theta^{(k)}/v^{(k)}r^{(k)2}\cos\left(\psi^{(k)} - \psi_p\right) \\ a_{51} = -L^{(k)}\cos\beta^{(k)}\left(1 - R_e r^{(k)}/H_0\right)/\left(v^{2(k)}\cos\theta^{(k)}\cos\left(\psi^{(k)} - \psi_p\right)\right) \\ \qquad -1/v^{(k)2}r^{(k)2}\cos\left(\psi^{(k)} - \psi_p\right) \\ a_{61} = -L^{(k)}\sin\beta^{(k)}\left(1 - R_e r^{(k)}/H_0\right)/\left(v^{2(k)}\cos\theta^{2(k)}\cos\left(\psi^{(k)} - \psi_p\right)\right) \end{cases} \tag{22}$$

$$\begin{cases} a_{23} = -\sin\psi^{(k)}\sin\varphi^{(k)}/\left(\cos^2\varphi^{(k)}\cos(\psi^{(k)} - \psi_p)\right) \\ a_{63} = -\sin\psi^{(k)}/\left(\cos^2\varphi^{(k)}\cos(\psi^{(k)} - \psi_p)\right) \end{cases} \tag{23}$$

$$\begin{cases} a_{44} = D^{(k)}r/\left(v^{2(k)}\cos\theta^{(k)}\cos(\psi^{(k)} - \psi_p)\right) \\ \qquad - \tan\theta^{(k)}/\left(v^{(k)2}r^{(k)}\cos(\psi^{(k)} - \psi_p)\right) \\ a_{54} = -2/\left(r^{(k)}v^{(k)3}\cos(\psi^{(k)} - \psi_p)\right) \end{cases} \tag{24}$$

$$\begin{cases} a_{15} = -r^{(k)} / \left(\cos^2 \theta^{(k)} \cos\left(\psi^{(k)} - \psi_p\right)\right) \\ a_{45} = D^{(k)} r^{(k)} \tan \theta^{(k)} / \left(v^{(k)} \cos \theta^{(k)} \cos\left(\psi^{(k)} - \psi_p\right)\right) \\ \qquad + 1 / \left(\cos^2 \theta^{(k)} v^{(k)} r^{(k)} \cos\left(\psi^{(k)} - \psi_p\right)\right) \\ a_{55} = -L^{(k)} \cos \beta^{(k)} r^{(k)} \tan \theta^{(k)} / \left(v^{2(k)} \cos \theta^{(k)} \cos\left(\psi^{(k)} - \psi_p\right)\right) \\ a_{65} = -2 L^{(k)} \sin \beta^{(k)} r^{(k)} \tan \theta^{(k)} / \left(v^{2(k)} \cos^2 \theta^{(k)} \cos\left(\psi^{(k)} - \psi_p\right)\right) \end{cases} \tag{25}$$

$$\begin{cases} a_{16} = -\tan \theta^{(k)} r^{(k)} \sin\left(\psi^{(k)} - \psi_p\right) / \cos^2\left(\psi^{(k)} - \psi_p\right) \\ a_{26} = \left(-\cos \psi^{(k)} \cos\left(\psi^{(k)} - \psi_p\right) - \sin \psi^{(k)} \sin\left(\psi^{(k)} - \psi_p\right)\right) / \left(\cos \varphi^{(k)} \cos^2\left(\psi^{(k)} - \psi_p\right)\right) \\ a_{36} = \left(\sin \psi^{(k)} \cos\left(\psi^{(k)} - \psi_p\right) - \cos \psi^{(k)} \sin\left(\psi^{(k)} - \psi_p\right)\right) / \cos^2\left(\psi^{(k)} - \psi_p\right) \\ a_{46} = D^{(k)} r^{(k)} \tan\left(\psi^{(k)} - \psi_p\right) / \left(v^{(k)} \cos \theta^{(k)} \cos\left(\psi^{(k)} - \psi_p\right)\right) \\ \qquad + \tan \theta^{(k)} \sin\left(\psi^{(k)} - \psi_p\right) / \left(v^{(k)} r^{(k)} \cos^2\left(\psi^{(k)} - \psi_p\right)\right) \\ a_{56} = -L^{(k)} \cos \beta^{(k)} r^{(k)} \tan\left(\psi^{(k)} - \psi_p\right) / \left(v^{2(k)} \cos \theta^{(k)} \cos\left(\psi^{(k)} - \psi_p\right)\right) \\ \qquad - \sin\left(\psi^{(k)} - \psi_p\right) / \cos^2\left(\psi^{(k)} - \psi_p\right) + \sin\left(\psi^{(k)} - \psi_p\right) / \left(r^{(k)} v^{(k)2} \cos^2\left(\psi^{(k)} - \psi_p\right)\right) \\ a_{66} = -L^{(k)} \sin \beta^{(k)} r^{(k)} \tan\left(\psi^{(k)} - \psi_p\right) / \left(v^{2(k)} \cos \theta^{2(k)} \cos\left(\psi^{(k)} - \psi_p\right)\right) \\ \qquad + \left(-\cos \psi^{(k)} \cos\left(\psi^{(k)} - \psi_p\right) \tan \varphi^{(k)} - \sin \psi^{(k)} \sin\left(\psi^{(k)} - \psi_p\right) \tan \varphi^{(k)}\right) / \cos^2\left(\psi^{(k)} - \psi_p\right) \end{cases} \tag{26}$$

$$\begin{cases} a_{47} = \left(2 c_{d3} \alpha^{(k)} + c_{d2}\right) r^{(k)} q^{(k)} s / \left(m g_0 v^{(k)} \cos \theta^{(k)} \cos(\psi^{(k)} - \psi_p)\right) \\ a_{57} = -c_{l2} \cos \beta^{(k)} r^{(k)} q^{(k)} s / \left(m g_0 v^{2(k)} \cos \theta^{(k)} \cos(\psi^{(k)} - \psi_p)\right) \\ a_{67} = -c_{l2} \sin \beta^{(k)} r^{(k)} q^{(k)} s / \left(m g_0 v^{2(k)} \cos \theta^{2(k)} \cos(\psi^{(k)} - \psi_p)\right) \end{cases} \tag{27}$$

$$\begin{cases} a_{58} = L^{(k)} \sin \beta^{(k)} r^{(k)} / \left(v^{2(k)} \cos \theta^{(k)} \cos(\psi^{(k)} - \psi_p)\right) \\ a_{68} = -L^{(k)} \cos \beta^{(k)} r^{(k)} / \left(v^{2(k)} \cos \theta^{2(k)} \cos(\psi^{(k)} - \psi_p)\right) \end{cases} \tag{28}$$

*3.2. Constraint Convexity and Relaxation*

(1)　Constraints of state variables

The model needs to establish certain trust region constraints on the latest iterative trajectory to ensure that the deviation is not too large, which can reduce the error between the convex model and the real model.

$$\left\{ \left| X - X^{(k)} \right| \leq \xi_X \right. \tag{29}$$

where $\xi_X$ is the trust region of state.

(2)　Convexification of process constraint

If the process constraint (8) of the re-entry glide vehicle is expanded on the R-V (H-V) profile

$$\begin{cases} r \geq \left(2 * H_0 \ln\left(\rho_0 (v * vc)^{3.15} C / \dot{Q}_{\max}\right) + R_e\right) / R_e = r_{\dot{Q}} \\ r \geq \left(H_0 \ln\left(\rho_0 (v * vc)^2 / 2 q_{\max}\right) + R_e\right) / R_e = r_q \\ r \geq \left(H_0 \ln\left(S \rho_0 (v * vc)^2 \sqrt{C_D^2 + C_L^2} / (2 n_{\max} m g_0)\right) + R_e\right) / R_e = r_n \end{cases} \tag{30}$$

where, $r_{\dot{Q}}, r_q, r_n$ represents thermal density, dynamic pressure and overload height.

Expand Equation (30) in the last iterative trajectory

$$
\begin{cases}
r_{\dot{Q}} = r_{\dot{Q}}^{(k)} + \left.\dfrac{\partial r_{\dot{Q}}}{\partial v}\right|_{X^{(k)}} \left(v - v^{(k)}\right) \\[2mm]
r_q = r_q^{(k)} + \left.\dfrac{\partial r_q}{\partial v}\right|_{X^{(k)}} \left(v - v^{(k)}\right) \\[2mm]
r_n = r_n^{(k)} + \left.\dfrac{\partial r_n}{\partial v}\right|_{X^{(k)}} \left(v - v^{(k)}\right) + \left.\dfrac{\partial r_{nj}}{\partial \alpha}\right|_{X^{(k)}} \left(u_3 - u_3^{(k)}\right)
\end{cases}
\tag{31}
$$

$$
\begin{cases}
\left.\dfrac{\partial r_{\dot{Q}}}{\partial v}\right|_{X^{(k)}} = 6.3 H_0 / \left(R_e v^{(k)}\right) \\[2mm]
\left.\dfrac{\partial r_q}{\partial v}\right|_{X^{(k)}} = 2 H_0 / \left(R_e v^{(k)}\right) \\[2mm]
\left[ \left.\dfrac{\partial r_n}{\partial v}\right|_{x^{(k)}}, \left.\dfrac{\partial r_n}{\partial \alpha}\right|_{x^{(k)}} \right] = \left[ 2 H_0 / \left(R_e v^{(k)}\right), \dfrac{H_0}{R_e} \dfrac{C_D^{(k)}(2cd_3\alpha^{(k)}+cd_2)+C_L^{(k)}cl_2}{C_D^{2(k)}+C_L^{2(k)}} \right]
\end{cases}
\tag{32}
$$

where $H_0$ represents the air density calculation constant.

(3)　　Convexification of the no-fly zones constraint

The no-fly zones constraint (9) is non-convex and cannot be solved directly and effectively, so it needs to be convex.

$$
\begin{aligned}
&\left(\phi^{(k)} - \phi_{bi}\right)^2 + \left(\varphi^{(k)} - \phi_{bi}\right)^2 + 2\left(\phi^{(k)} - \phi_{bi}\right)\left(\phi - \phi^{(k)}\right) \\
&+ 2\left(\varphi^{(k)} - \varphi_{bi}\right)\left(\varphi - \varphi^{(k)}\right) - r_{bi}^2 \geq 0
\end{aligned}
\tag{33}
$$

### 3.3. Improved Description of Convex Optimization Problem

According to the settings in Sections 3.1 and 3.2, the new optimization problem can be transformed into

$$
J_1 = J_0 - C_3 \int_{S_f}^{0} |\dot{\alpha}| + \left|\dot{\beta}\right| dS
\tag{34}
$$

Then convert problem $P_0$ into problem $P_1$ through convexity.

$$
\begin{aligned}
P_1 : \min \quad & J_1 \\
\text{subject to}: \quad & (20), (29), (31), (33) \\
& X\left(S_{f0}\right) = [r_0, \phi_0, \varphi_0, v_0, \theta_0, \psi_0], \; v(s=0) = v_f
\end{aligned}
\tag{35}
$$

### 3.4. Discretization of Dynamic Model

The problem $P_1$ is a representation in the continuous flight range domain, which is a model with infinite dimension. In order to further solve the problem, it needs to be discretized. First, set grid point locations of the generated trajectory in the flight range domain as $S_i$, $i = 1, \ldots, m$ represents the serial number of the grid point, $m$ represents the number of the grid points, $S_1 = S_{f0} \geq S_{i+1} \geq S_i \geq S_m = 0$.

Then the dynamic Equation (20) can be discretized using the Euler method.

$$
X_{i+1} = X_i + \Delta S_i \dot{X}_i = X_i + \Delta S_i \left(A_i^{(k)} X_i + B_i^{(k)} U_i + C_i^{(k)}\right)
\tag{36}
$$

where, $\Delta S_i$ represents the projected range-to-go difference between grid point $i$ and $i + 1$.

At the same time, due to different iterations, the trust region size required of optimization may be different, so the variable trust region method is adopted in Equation (30) to improve the search range of solution

$$
\left\{ \left|X_j - X_j^{(k)}\right| \leq \eta \xi_x \right.
\tag{37}
$$

where $\eta$ represents variable of trust region

On this basis, the optimization objective $J_1$ is further discretized to obtain a new optimization objective $J_2$

$$J_2 = J_0 - \kappa_3 \left( \sum_{j=0}^{N} |\dot{\alpha}_j| \Delta S_j + \sum_{j=0}^{N} |\dot{\beta}_j| \Delta S_j \right) + \kappa_4 \eta \tag{38}$$

where $\kappa_3$, $\kappa_4$ are optimization coefficients.

Then discretize the constraints (31), (33) to further obtain the final optimization problem $P_2$

$$\begin{aligned} P_2 : \min & \quad J_2 \\ subject\ to: & \quad (31), (32), (36), (37) \\ & \quad X\left(S_{f0}\right) = [r_0, \phi_0, \varphi_0, v_0, \theta_0, \psi_0], \ v(s=0) = v_f \end{aligned} \tag{39}$$

*3.5. Termination Condition of Solving*

On the basis of problem $P_2$, when the convexity value of each grid point for two consecutive times iteration meets the convergence threshold $\varepsilon$, the convexity iteration is terminated. The iteration termination conditions are as follows

$$\begin{cases} \max_{1 \leq i \leq n} \left| r^{(k+1)}(S_i) - r^{(k)}(S_i) \right| \leq \varepsilon_r, \ \max_{1 \leq i \leq n} \left| \phi^{(k+1)}(S_i) - \phi^{(k)}(S_i) \right| \leq \varepsilon_\phi \\ \max_{1 \leq i \leq n} \left| \varphi^{(k+1)}(S_i) - \varphi^{(k)}(S_i) \right| \leq \varepsilon_\varphi, \ \max_{1 \leq i \leq n} \left| v^{(k+1)}(S_i) - v^{(k)}(S_i) \right| \leq \varepsilon_v \\ \max_{1 \leq i \leq n} \left| \theta^{(k+1)}(S_i) - \theta^{(k)}(S_i) \right| \leq \varepsilon_\theta, \ \max_{1 \leq i \leq n} \left| \psi^{(k+1)}(S_i) - \psi^{(k)}(S_i) \right| \leq \varepsilon_\psi \\ \max_{1 \leq i \leq n} \left| \alpha^{(k+1)}(S_i) - \alpha^{(k)}(S_j) \right| \leq \varepsilon_\alpha, \ \max_{1 \leq i \leq n} \left| \beta^{(k+1)}(S_i) - \beta^{(k)}(S_i) \right| \leq \varepsilon_\beta \end{cases} \tag{40}$$

where $\left[ \varepsilon_r, \varepsilon_\phi, \varepsilon_\varphi, \varepsilon_v, \varepsilon_\theta, \varepsilon_\psi, \varepsilon_\alpha, \varepsilon_\beta \right]$ represent the convergence threshold of the corresponding state and control variables.

## 4. Fast Initial Trajectory Setting and Distributed Grid Points Adjustment

*4.1. Fast Initial Trajectory Setting*

In order to improve the solution seeking speed of the method and ensure that the initial trajectory meets a large number of constraints as much as possible, according to the landing points of the vehicle, a fast initial trajectory setting method that meets the dynamic models was designed. Since the terms of Coriolis acceleration and implicated acceleration caused by the earth rotation are small, the flight path of the re-entry glide vehicle in any direction, at any longitude and latitude coordinate can be approximated to the equatorial origin, and with the same initial speed and same control law in the longitudinal direction by rotating and moving, which is shown in Figure 2a,b. In this paper, the landing points map of the constant angle of attack and bank angle control strategy of the re-entry glide vehicle is established offline, and suitable control variables and initial mapping trajectory are obtained by interpolation, as shown in Figure 2b. Finally, the initial generated trajectory satisfying the constraint can be obtained by integrating the corresponding control law in the original state.

Reachable landing point locations of the re-entry glide vehicle are shown in Figure 3a,b. Among them, the change of constant attack angle and constant bank angle has regularity. Generally, with the fixed attack angle and the different bank angle, the lateral landing points locations are mainly affected. With the same bank angle and different attack angle, the flight range is affected. When the setting of the attack angle is less than the maximum lift drag ratio attack angle, the flight range is proportional to the attack angle; when it is greater than the maximum lift drag ratio attack angle, the flight range is inversely proportional to the attack angle.

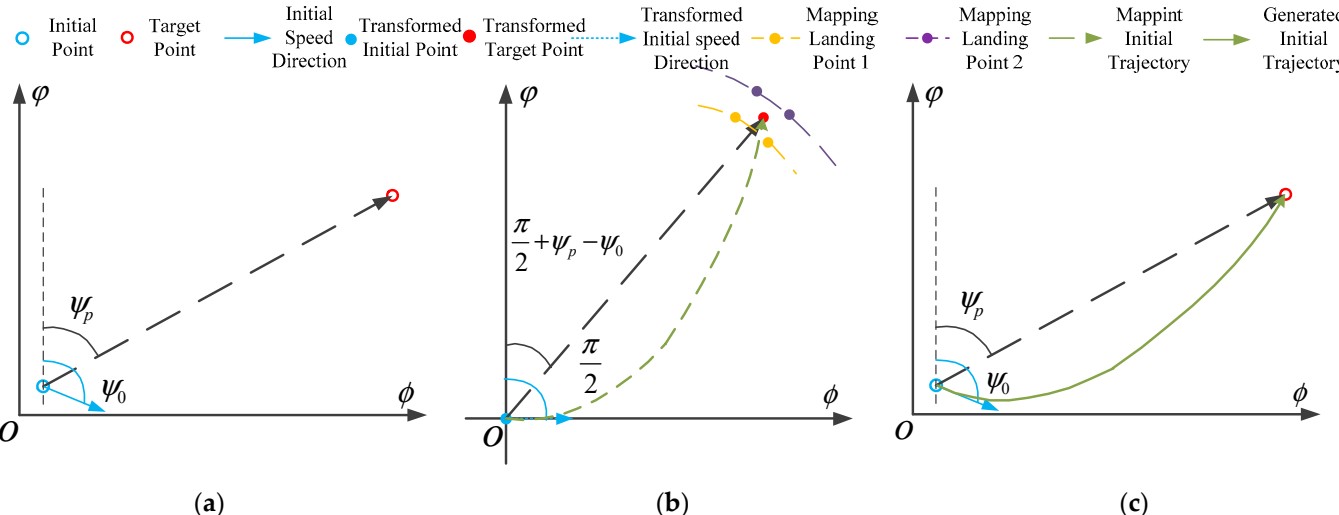

**Figure 2.** Fast initial trajectory setting method: (**a**) represents the initial setting relationship; (**b**) represents the transformed mapping relationship; (**c**) represents the generated initial trajectory.

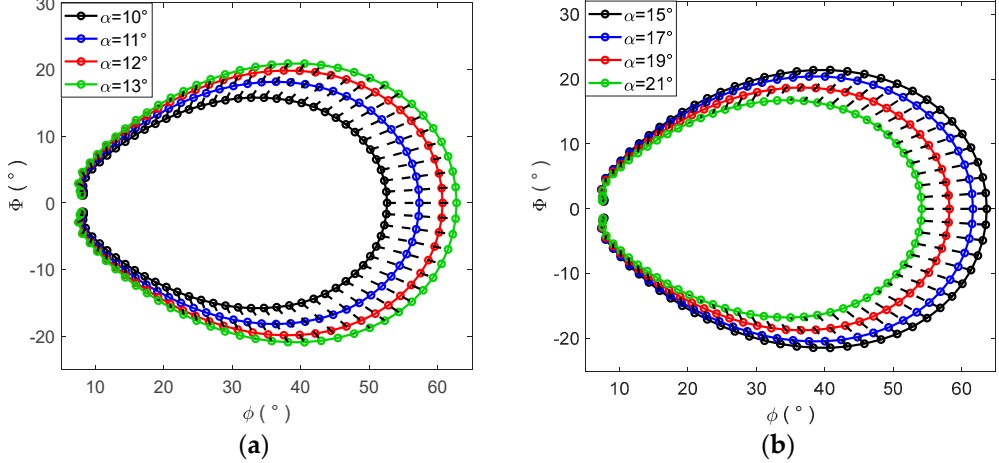

**Figure 3.** Fast initial trajectory setting method: (**a**) represents the change of landing points with a minor attack angle; (**b**) represents the change of landing points with a high attack angle.

### 4.2. Distributed Grid Points Adjustment

According to the literature [28], with the last iterative grid points, the grid points can be effectively adjusted by calculating the non-linear illegal degree of the grid points. The non-linear illegal degree of the grid point *i* is

$$\Delta n_i = \sum_{i=2}^{m} \left\| X^{(k)}(S_i) - \int_{S_{i-1}}^{S_i} F(X, U, S) dS \right\|_2 \tag{41}$$

Let the number of grid point intervals be $K$, $\Delta S_j = (S_i - S_{j-1})/K$

$$\int_{S_{i-1}}^{S_i} F(X, U, S) dS = \sum_{j=1}^{K} F(X, U, S_i + j\Delta s_j) \tag{42}$$

In order to adjust grid points more quickly and dynamically, this paper defines the estimated nonlinear illegal degree distribution probability density function $P$, and defines increasing the value threshold of the grid points' nonlinear illegal degree $\Delta n_{max}$. When the nonlinear illegal degree of a point exceeds this value, it means that the nonlinearity illegal degree of this point is high, and grid points need to be increased; At the same time, the

deleting value threshold $\Delta n_{\min}$ is also defined. The grid point supplement function $P'$ of a point is expressed as

$$P'(S_i) = \begin{cases} 0, & if \; \Delta n_i \leq \Delta n_{\min} \\ 1, & if \; \Delta n_{\min} \leq \Delta n_i \leq \Delta n_{\max} \; or \; i = 1 \; or \; i = n \\ \Delta n_i / \Delta n_{\max}, & if \; \Delta n_i \geq \Delta n_{\max} \end{cases} \tag{43}$$

At the same time, in order to ensure the iteration speed of the method, set the upper number of grid points $n_{\max}$, and the adjusted number of grid points $n^{(k+1)}$ meets

$$n^{(k+1)} = \begin{cases} \sum\limits_{i=1}^{n} P'(S_i), & if \; \sum\limits_{i=1}^{n} P'(S_i) \leq n_{\max} \\ n_{\max}, & if \; \sum\limits_{i=1}^{n} P'(S_i) > n_{\max} \end{cases} \tag{44}$$

The number of grid points to be adjusted is $\Delta n^{(k+1)} = Count(P' = 0) + n^{(k+1)} - n^{(k)}$.

Let the probability density function of the adjusted grid points be $P$, and it can be expressed as

$$P(S_i) = P'(S_i) / \left( \sum\limits_{i=1}^{n} P'(S_i)|\Delta S_i| \right) \tag{45}$$

For any distance domain location $S$, the probability density of this point can be obtained by interpolating by the calculated value of Equation (44). Finally, the corresponding cumulative distribution function is inversely solved by (45) to obtain the location, when the corresponding cumulative probability value is equal to $i/\left(\Delta n^{(k+1)} + 1\right), i = 1, \dots, n^{(k+1)}$ as the additional grid point location.

As shown in Figure 4, it is assumed that after the generation of initial trajectory, the grid points will be a uniform distribution according to the flight range domain in the first iteration. After the first iteration, the probability density function distribution of grid points and the degree of nonlinear illegal degree will be calculated. In the first iteration, the red point location is of lower nonlinear illegal degree, so it can be deleted for adjustment. According to the probability density function distribution, the adjustment position is the blue dotted point of the second iteration; After the second optimization, find the location where the degree of nonlinear illegal is lower again, and adjust them to the new grid points location according to the second probability distribution, and continue to the third iteration.

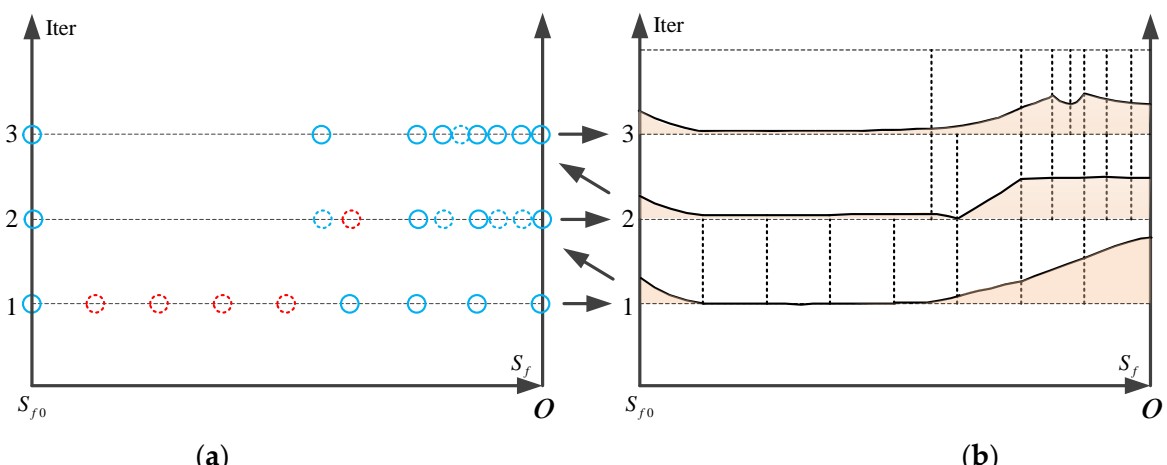

**Figure 4.** Grid points adjustment method: (**a**) represents the location of grid points; (**b**) represents the distribution of the probability density function distribution of grid points.

## 5. Simulation

### 5.1. Experimental Subjects and Parameter Settings

In this paper, CAV-H is selected as the simulation vehicle, and its specific parameters can be referred to in the literature [29], where its process constraints are set $\dot{Q}_{\max} = 4 \times 10^6$ W/m$^2$, $q_{\max} = 100$ Kpa, $n_{\max} = 4$. The trust region $[\xi_r, \xi_\phi, \xi_\varphi, \xi_v, \xi_\theta, \xi_\psi, \xi_\alpha, \xi_\beta]$ are $[2 \times 10^4 / R_e, 500 \times 10^4 \pi / R_e, 500 \times 10^4 \pi / R_e, 1000 / v_c, 20\pi / 180, 30\pi / 180, 10\pi / 180, 90\pi / 180]$, convergence error threshold $[\varepsilon_r, \varepsilon_\phi, \varepsilon_\varphi, \varepsilon_v, \varepsilon_\theta, \varepsilon_\psi, \varepsilon_\alpha, \varepsilon_\beta]$ are $[200 / R_e, 5 \times 10^4 \pi / R_e, 5 \times 10^4 \pi / R_e, 100 / v_c, 1\pi / 180, 5\pi / 180, 1\pi / 180, 5\pi / 180]$, maximum number of grid points $n_{\max} = 200$. The simulation software is python with Win10, deploying ECOS-BB solver for sequential programming and Matlab for plotting. The simulation CPU is Inter-I7.

### 5.2. Verification of the Initial Trajectory Generation

In order to verify the effectiveness of the initial trajectory generation method proposed in this paper, the three types of relationship between initial point and target point are selected to be tested, respectively, to analyze the guidance effects and compare with the iterative trajectories. Where the initial states of all trajectories are set as $[r_0, \phi_0, \varphi_0, v_0, \theta_0, \psi_0] = [(R_e + 70 \text{ km}) / R_e, 0°, 0°, 6000 / v_c, -2°, 45°]$. Then, three guidance location are selected to generate trajectories on different sides in the velocity direction of the initial state $[r_{f1}, \phi_{f1}, \varphi_{f1}] = [(R_e + 30 \text{ km}) / R_e, 30°, 15°]$, $[r_{f2}, \phi_{f2}, \varphi_{f2}] = [(R_e + 30 \text{ km}) / R_e, 34°, 24°]$, $[r_{f3}, \phi_{f3}, \varphi_{f3}] = [(R_e + 30 \text{ km}) / R_e, 24°, 34°]$. It can be seen from Figure 5a,b that the longitudinal and lateral landing point of the initial trajectory can basically reach the target point by the initial trajectory generation algorithm in this paper. Then, target point 2 was taken as the example to compare the iterative trajectory. As can be seen from Figure 5c,d, by our method the vehicle could effectively converge to the optimal trajectory after solving for five iterations with a total time of 18.46 s. When the maximum attack angle strategy method was adopted as the initial trajectory in this paper, the number of iterations of the algorithm increased to nine. The total time is 40.21 s, which indicates our initial trajectory generation method has a better effectiveness for the optimization algorithm.

### 5.3. Validity Verification

In order to verify the effective generation of the re-entry trajectory under constraints by the proposed method, three cases are selected for verification. The initial states of the vehicle are the same in all three cases. $[r_0, \phi_0, \varphi_0, v_0, \theta_0, \psi_0] = [(R_e + 70 \text{ km}) / R_e, 0°, 0°, 6000 / v_c, 0°, 60°]$ when the terminal states are also the same $[r_f, \phi_f, \varphi_f, v_f] = [(R_e + 30 \text{ km}) / R_e, 30°, 25°, 2000 / v_c]$. In the three cases, the vehicle will have a flight with the no no-fly zone, flying- around the no-fly zones and passing through the no-fly zones in three cases. In case 2,3, the no-fly zones are set as follows:

$$\text{Case 2}: [\phi_{b1}, \varphi_{b1}, r_{b1}] = [15°, 8.5°, 2°], \ [\phi_{b1}, \varphi_{b1}, r_{b1}] = [24°, 16°, 2°]$$

$$\text{Case 3}: [\phi_{b1}, \varphi_{b1}, r_{b1}] = [15°, 11°, 2°], \ [\phi_{b1}, \varphi_{b1}, r_{b1}] = [24°, 14°, 2°]$$

The simulation effect is shown in Figures 6–8. As can be seen from Figure 6, the convexation algorithm proposed in this paper can realize effective trajectory generation after a variety of situations for all three cases. As can be seen from case 1, although the longitudinal guidance effect of the initial trajectory is not very good, the method can realize accurate lateral and longitudinal trajectory generation. On the other hand, it can be seen from cases 2 and 3 that although there is crossover between the initial trajectory and the no-fly zones, the method can avoid the no-fly zone through effective adjustment and satisfy the longitudinal guidance at the same time.

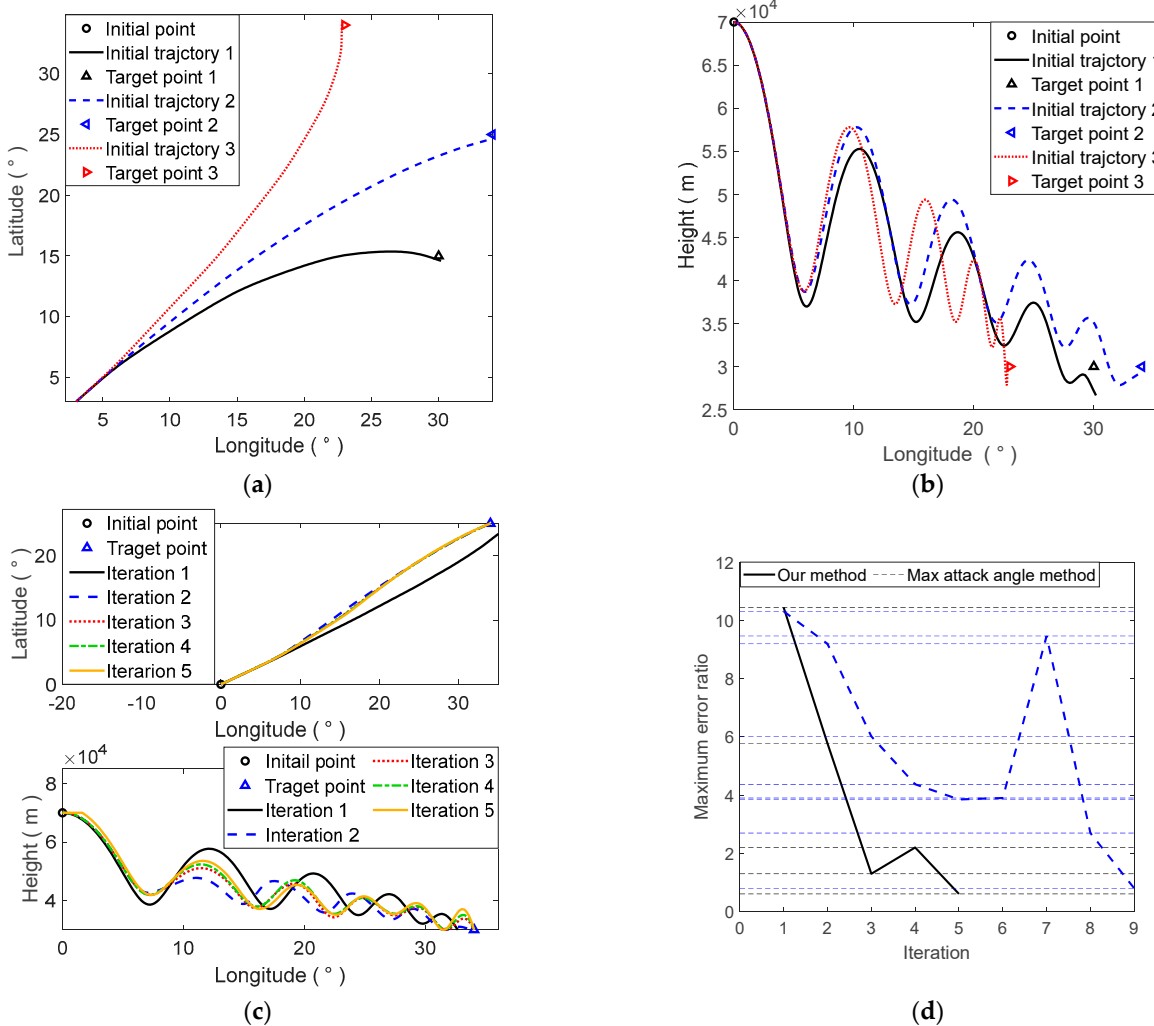

**Figure 5.** The generated initial trajectories: (**a**) is the lateral effect of trajectory; (**b**) is the longitudinal effect; (**c**) is the iteration change of the optimized trajectories; (**d**) is the comparison between our method and max attack angle method.

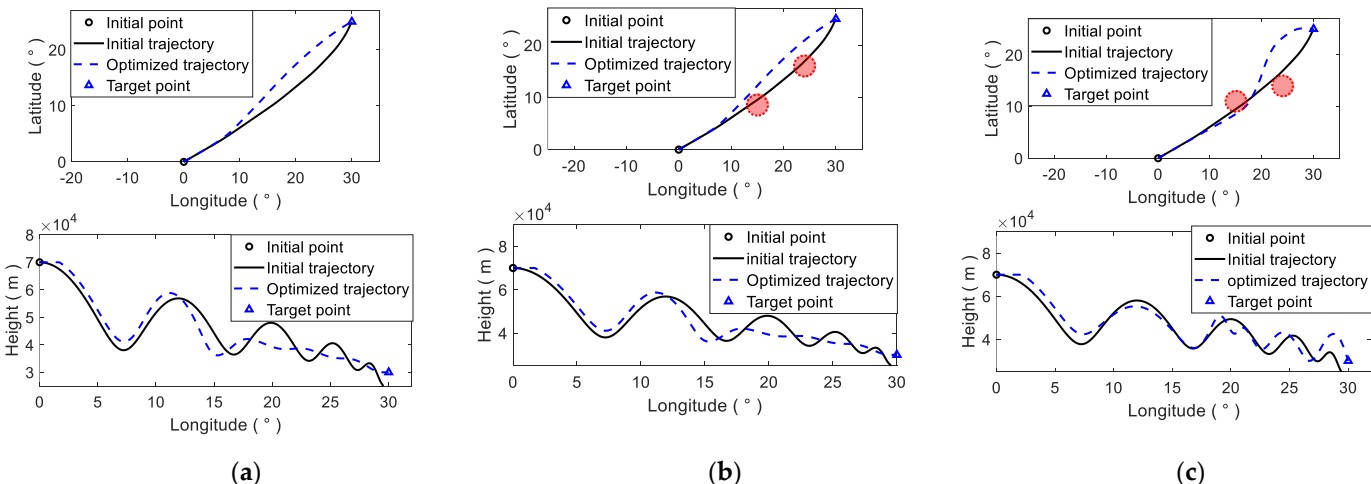

**Figure 6.** The generated optimized trajectories of cases: (**a**) is the longitudinal and lateral trajectories of case 1; (**b**) is the longitudinal and lateral trajectories of case 2; (**c**) is the longitudinal and lateral trajectories of case 3.

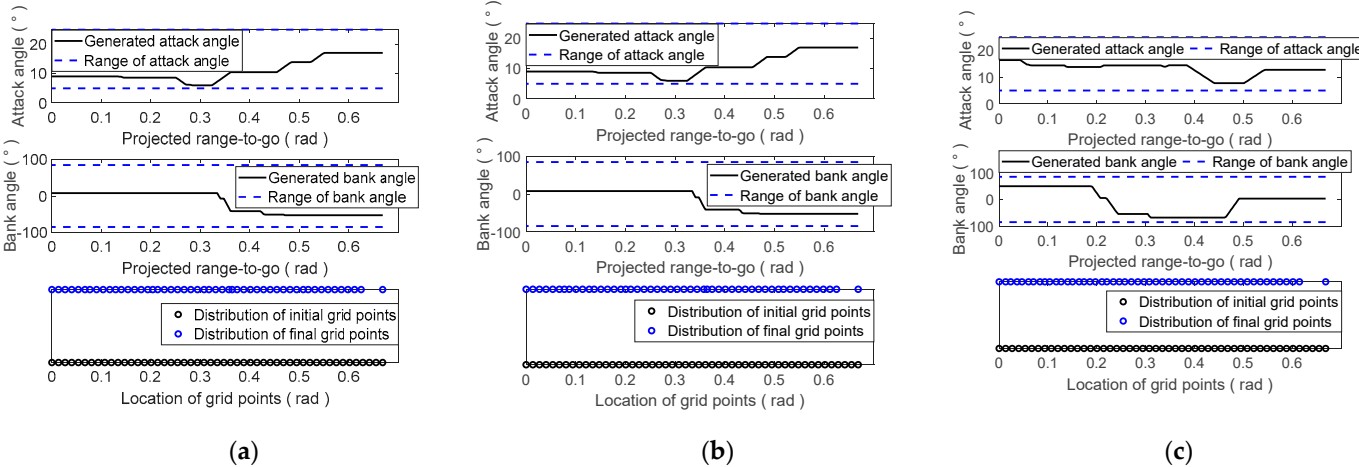

**Figure 7.** The grid points and control angle: (**a**–**c**) are the distributions of the grid points and control angles of case 1, 2 and 3, respectively.

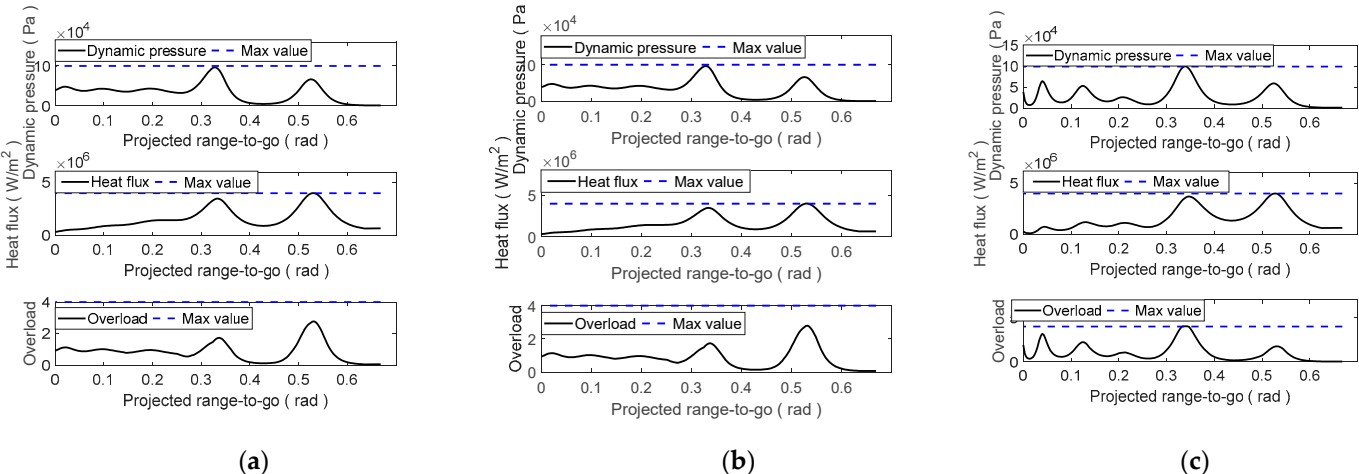

**Figure 8.** The processing constraints of cases: (**a**–**c**) are the processing constraints of case 1, 2 and 3, respectively.

On the other hand, according to the effect of Figures 6–8, all the trajectories can meet the final constraint requirements. In Case 2 and Case 3, the peak values of several constraints are high due to the avoidance of the no-fly zones, especially at low flight altitudes. In order to effectively meet the constraints, at the initial stage of flight, the vehicle usually employs a high attack angle. In case 3, in order to pass through and between the no-fly zones and increase the range, a smaller attack angle value is adopted; The change of the corresponding bank angle represents the avoidance maneuver process of the vehicle. In Case 1, there are no no-fly zones, so the change of the bank angle is small; However, in Case 2 and Case 3, considering the influence of no-fly zones, there are many variation of the bank angle; this can be seen from the grid point changes when the uniform grid points are used in the initial trajectory. For the last iteration trajectory, due to the small aerodynamic force in the initial stage of vehicle, where the nonlinear illegal degree is low, and the grid points are relatively sparse; at the end of flight, the flight height is low, the aerodynamic force changes greatly, and the grid points are obviously dense.

The optimized trajectories' accuracy, iteration times and the waste time of the three cases are shown in Table 1. Where $E_l$ represents the lateral error and $E_h$ represents the longitudinal error. There is a no-fly zone in Case 1, so its lateral and longitudinal trajectories generation accuracy is very high; due to the influence of the no-fly zone, Case 2 and Case 3 have poor guidance accuracy, but the lateral error is not more than 1.6 km, and the

longitudinal error is not more than 100 m. For the number of iteration convergence, Case 3 needs to pass through and between the no-fly zones, so it has many iterations and takes a longer time, but the average iteration time is not more than 6 s.

**Table 1.** Generation effect with different cases.

| Case | $E_l$ (km) | $E_h$ (km) | Waste Time (s) | Iteration |
|------|-----------|-----------|----------------|-----------|
| 1 | 0.001 | $-6.43 \times 10^{-4}$ | 22.93 | 5 |
| 2 | 0.041 | 0.005 | 27.20 | 5 |
| 3 | 0.016 | −0.001 | 110.67 | 19 |

### 5.4. Simulation Experiment with Different Target Points

The simulations are operated with the same initial and final states and different no-fly zones. The simulated results of different target points are analyzed in this section. The target points and simulation results are shown in Table 2. The setting of the no-fly zones is the same as that in Case 3 in Section 5.3. The simulation effect is shown in Figure 9. It can be seen from Figure 9 that for different target points, this method can realize lateral and longitudinal guidance with a high accuracy. When effective avoidance is achieved, reasonable control angles are adopted. Among them, a trajectory with a long range and requiring the avoidance of no-fly zones has more iterations, relatively poor accuracy and more frequent changes in control angles.

**Table 2.** Generation effect with different target points.

| Case | $\phi_f$ (°) | $\varphi_f$ (°) | $E_l$ (km) | $E_h$ (km) | Waste Time (s) | Iteration |
|------|-------------|----------------|-----------|-----------|----------------|-----------|
| 1 | 28 | 23 | 0.167 | 0.009 | 135.84 | 23 |
| 2 | 30 | 20 | 0.082 | −0.002 | 38.63 | 7 |
| 3 | 30 | 10 | 0.079 | −0.009 | 34.87 | 6 |
| 4 | 27 | 27 | 0.139 | −0.019 | 122.70 | 21 |

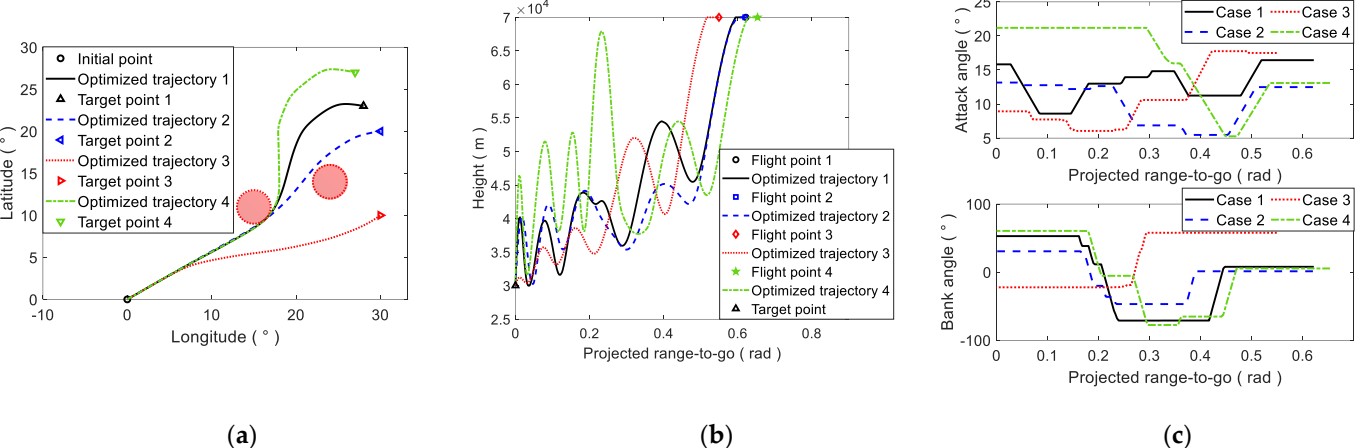

(**a**)　　　　　　　　　　　　(**b**)　　　　　　　　　　　　(**c**)

**Figure 9.** The generation effects of different target points: (**a**) represents the lateral trajectories; (**b**) represents the longitudinal trajectories; (**c**) represents the control angles.

### 5.5. Simulation Experiment with Initial State Disturbance

To further demonstrate the effectiveness and robustness of our method, the simulation is operated when the initial state of the vehicle is disturbed with the same target point and the no-fly zones. The target point and no-fly zones are set as the same as Case 2 in Section 5.3, the initial altitude $\Delta h_0 = +2$ km, initial speed $\Delta v_0 = -100$ m/s, initial flight path angle $\Delta \theta_0 = +0.5°$ and initial flight heading angle $\Delta \psi_0 = -10°$ are, respectively, simulated. The optimization effect is shown in Figure 10 and Table 3. It can be seen from

Figure 10 that with certain disturbance of the initial state, the vehicle can still effectively avoid the no-fly zone and reach the target points with high accuracy in both the lateral and longitudinal directions. The control angle obtained by changing the height, speed and the flight angle is similar, which has a certain impact on the longitudinal trajectory. When the initial flight heading angle is changed, the optimization trajectory changes obviously, and the number of iterations in this case increases significantly. This is because the adjusted initial flight heading angle is larger than the LOS angle. In general, the proposed method is robust to initial state disturbances.

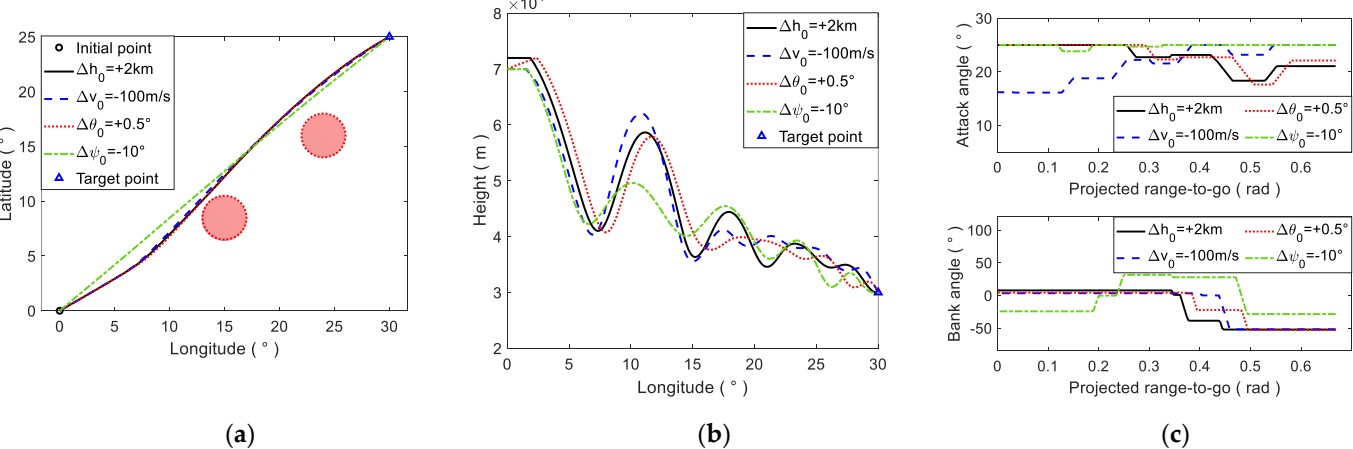

**Figure 10.** The generation effects of the initial state with disturbance: (**a**) represents the lateral trajectories; (**b**) represents the longitudinal trajectories; (**c**) represents the control angles.

**Table 3.** Generation effect with disturbance of the initial state.

| Case | $E_l$ (km) | $E_h$ (km) | Waste Time (s) | Iteration |
|---|---|---|---|---|
| $\Delta h_0 = +2$ km | 0.002 | $-0.001$ | 24.92 | 5 |
| $\Delta v_0 = -100$ m/s | 0.005 | $-9 \times 10^{-4}$ | 19.63 | 4 |
| $\Delta \theta_0 = +0.5$º | 0.002 | $-0.002$ | 25.08 | 5 |
| $\Delta \psi_0 = -10$º | 0.004 | $-6 \times 10^{-4}$ | 45.54 | 9 |

### 5.6. Monte Carlo Robustness Simulation

In order to further reflect the robust effect of the method proposed in this paper on process disturbance, Monte Carlo simulations are operated 100 times with the same conditions of Case 2 in Section 5.2, where the lift and drag coefficient, vehicle mass and reference area have 5% random error each time, and the atmospheric density has 20% random error, which are used to verify the robustness of our method. The results are shown in Figure 11. As can be seen in Figure 11, the convexity trajectory generation algorithm proposed in this paper is very accurate in lateral and longitudinal accuracy in the Monte Carlo test, which proves the robustness of our method with process disturbance conditions.

### 5.7. Comparison of Mainstream Methods

In order to reflect the fast-planning speed of the method proposed in this paper, the method in this paper is compared with the Gauss pseudo spectral method. The optimization results of different methods are shown in Figure 12 and Table 4. The trajectory obtained by using the Gauss pseudo spectral method, and the method in this paper, can effectively realize trajectory generation, and meet various constraints when the accuracy of the method is comparable. The trajectory generated by the Gauss pseudo spectral method is shorter, but the control amount changed greatly. In terms of the number of grid points, the method in this paper uses 200 grid points due to the upper limit of grid points, while the Gaussian pseudo spectral method finally uses 341 grid points; in terms of planning time, the method

in this paper takes 110.67 s, the Gauss pseudo spectral method takes 154.2 s, and the programing time is longer.

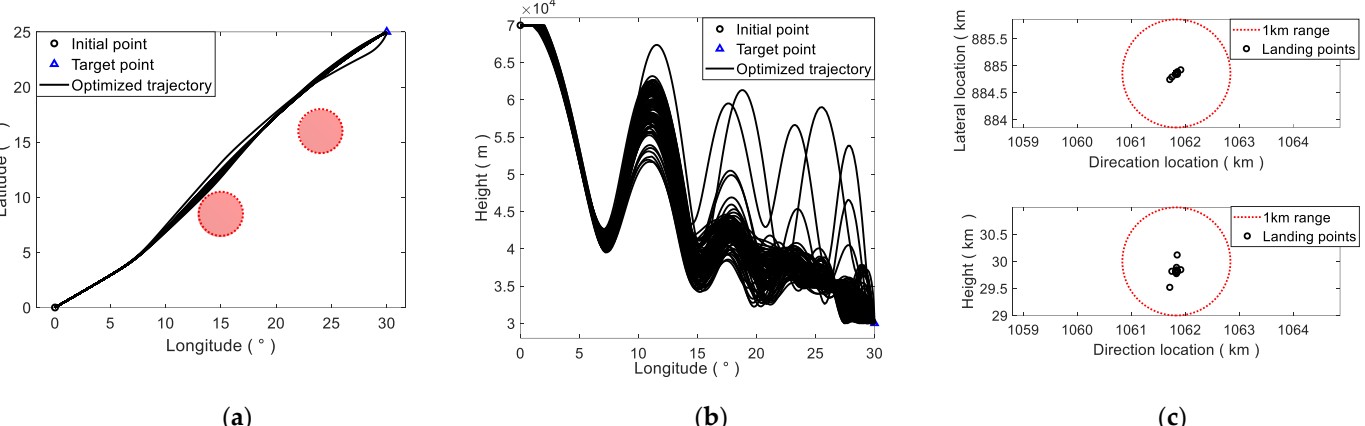

**(a)**        **(b)**        **(c)**

**Figure 11.** The generation effects of Monte Carlo experiments: (**a**) represents the lateral trajectories; (**b**) represents the longitudinal trajectories; (**c**) represents the landing points location.

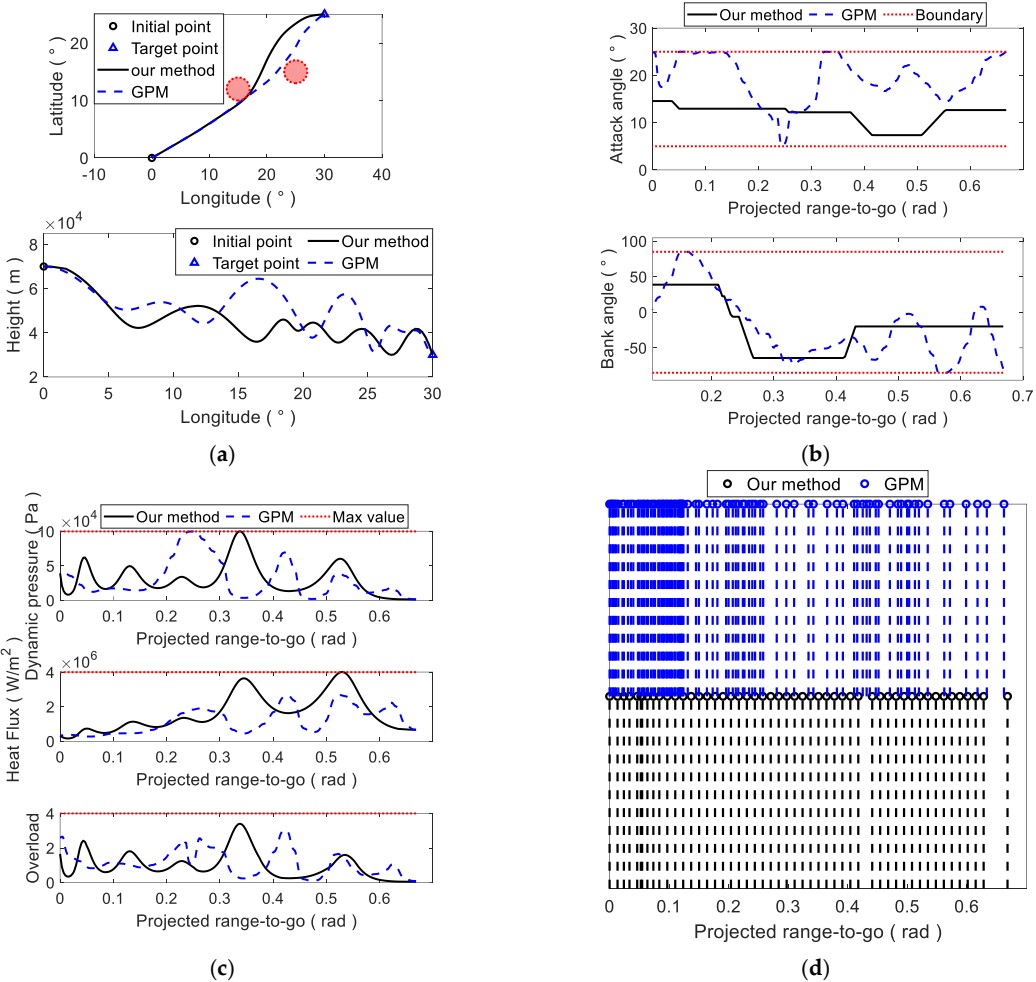

**(a)**        **(b)**

**(c)**        **(d)**

**Figure 12.** The comparison with GPM: (**a**) represents the lateral and longitudinal trajectories; (**b**) represents the control angle; (**c**) represents the processing constraints; (**d**) represents the grid points of the methods.

**Table 4.** Generation effect of different methods.

| Method | $E_l$ (km) | $E_h$ (km) | Waste Time (s) | Iteration | Number of Grid Points |
|---|---|---|---|---|---|
| Our method | 0.016 | 0.001 | 110.67 | 21 | 200 |
| Gaussian Spectral method | 0.010 | $1 \times 10^{-4}$ | 154.20 | 25 | 341 |

Through several simulations in this section, the effectiveness and robustness of our method have been verified. But the method of changing the maximum number of gird points adaptively could be researched in the future. Inspired by the multi-agent system guidance and control technology [30,31], convex trajectory generation algorithm of multi-re-entry glide targets will also be researched in our future work.

## 6. Conclusions

In this paper, according to the guidance mechanism of the vehicle, the concept of the projected range-to-go of the vehicle is first proposed. Then, the dynamic model is converted to the flight range domain when the model is convexated and discretized. In order to improve the generation speed and accuracy, a fast and accurate initial trajectory-generating method is proposed according to the landing points of the vehicle under different control laws. According to the nonlinear illegal degree of iterative trajectory, a grid point probability density function is proposed to dynamically change grid points. Through the simulation experiments against various disturbances, all of our final guidance errors are less than 1 km, and waste times are less than 135.62 s, which prove the effectiveness and robustness of our method. Compared with the conventional Gaussian pseudo-spectral method, we can obtain comparable accuracy and a faster generation speed.

**Author Contributions:** Conceptualization, M.L. and C.Z.; methodology, M.L. and C.Z.; software, M.L.; validation, M.L. and C.L.; writing—original draft preparation, M.L. and C.Z.; writing—review and editing, L.S. and H.L.; supervision, H.L. All authors have read and agreed to the published version of the manuscript.

**Funding:** This work was supported by the National Natural Science Foundation of China (Grant no. 62173339).

**Institutional Review Board Statement:** No applicable.

**Informed Consent Statement:** No applicable.

**Data Availability Statement:** All data generated or analyzed during this study are included in this published article.

**Conflicts of Interest:** The authors declare no conflict of interest.

## Nomenclature

| Magnitude | Meaning | Units |
|---|---|---|
| $r$ | Dimensionless geocentric distance of RGV | |
| $\phi$, $\varphi$ | Longitude and Latitude of RGV | rad |
| $v$ | Dimensionless velocity of RGV | |
| $R_e$ | Radius of earth | m |
| $\theta$ | Flight path angle of RGV | rad |
| $\psi$ | Flight heading angle of RGV | rad |
| $L, D$ | Dimensionless lift and drag force of HGV | |
| $\rho_0$ | Atmospheric density constant | kg/m$^3$ |
| $\rho$ | Atmosphere density constant | kg/m$^3$ |
| $H_0$ | Atmospheric altitude constant | m |
| $C_L, C_D$ | Lift coefficient and Drag coefficient | |
| $v_c$ | Dimensional velocity | m/s |

| $g_0$ | Gravity acceleration at zero altitude | m/s$^2$ |
|---|---|---|
| $\psi_p$ | Flight heading angle of the target point relative to the initial point | rad |
| $s$ | Projected range-to-go | rad |
| $r_f, v_f$ | Dimensionless terminal altitude | |
| $v_f$ | Dimensionless terminal velocity | |
| $\phi_f, \varphi_f$ | Terminal longitude and latitude | |
| $Q$ | Hear flux | W/m$^2$ |
| $q$ | Dynamic pressure | Pa |
| $n$ | Overload | |
| $\alpha$ | Attack angle | rad |
| $\beta$ | Bank angle | rad |
| $J$ | Objective function | |
| $\eta$ | Variable of trust region | |
| $P$ | Probability density function of grid points | |

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
