# Peer review of "A Trajectory Generation Algorithm for a Re-Entry Gliding Vehicle Based on Convex Optimization in the Flight Range Domain and Distributed Grid Points Adjustment"

_applsci, doi:10.3390/app13031988_

Round 1
Reviewer 1 Report
The presentation of this work should be redone to highlight the novelty better. It is this reviewer's opinion that the novelty (and the validation of the approach) is not focused on because the authors are attempting too much content for a single paper. Not enough focus was given to the methodology.
The innovation of the paper list three contributions; each could be its own paper. Because all three are included in a single paper, none of the three could be properly explored. As a result, most sections are unclear; the sections that are clear are not validated.
Other issues:
-Several terms were never defined,
-The authors refer to "the literature" in reference to "nonlinear illegal degree" but provide no citation and the explanation of this unfamiliar method was not clear.
-a low fidelity, but convex model was generated but no validation was shown (only verification).
-The conclusions section is one massive run-on sentence.
Author Response
Dear reviewer:
Thank you for your letter and the comments concerning our manuscript entitled “A trajectory generation algorithm for reentry gliding vehicle based on convex optimization in the flight range domain and distributed grid points adjustment” (ID: appisci 2169933). Those comments are all valuable and helpful for revising and improving our papers, as well as the important guiding significance to our researches. We have studied comments carefully and have made corrections which we hope meeting with approval. The main corrections in the paper and the responds to the comments are as follows:
Q1: The presentation of this work should be redone to highlight the novelty better. It is this reviewer's opinion that the novelty (and the validation of the approach) is not focused on because the authors are attempting too much content for a single paper. Not enough focus was given to the methodology. The innovation of the paper list three contributions; each could be its own paper. Because all three are included in a single paper, none of the three could be properly explored. As a result, most sections are unclear; the sections that are clear are not validated.
Reply: According to the opinion of the reviewer, we have reedited our manuscript, to highlight the innovation. But I think the trajectory generation convex model of reentry glide vehicle in the flight range domain is the base of manuscript, and the initial trajectory generation method and distributed grid points are the import improvement of the trajectory generation. All of whole are designed to generate the optimized trajectory better. We hope to keep them all in our manuscript.
Q2: Several terms were never defined,
Reply: We have checked all the manuscript,and all the current terms are defined now.
Q3: The authors refer to "the literature" in reference to "nonlinear illegal degree" but provide no citation and the explanation of this unfamiliar method was not clear.
Reply: The degree of nonlinearity is defined in reference [25], which has been marked in yellow in our manuscript.
Q4: a low fidelity, but convex model was generated but no validation was shown (only verification).
Reply: The dynamic model and constraints are linearized, and the Hessen matrix of the objective function is also non-positive definite, indicating that the convexation of vehicle trajectory is completed, and the range of iteration is gradually converging to the termination condition. Several cases are also listed to illustrate the effectiveness of the method.
Q5: The conclusions section is one massive run-on sentence.
Reply: The conclusion section has been rewritten as follow:
In this paper, according to the guidance mechanism of the vehicle, the concept of the projected range-to-go of the vehicle is proposed firstly. Then the dynamic model is con-verted to the flight range domain when the model is convexation and discretization. In order to improve the generation speed and accuracy, a fast and accurate initial trajectory generating method is proposed according to the landing points of the vehicle under different control laws. According to the nonlinear illegal degree of iterative trajectory, a grid point probability density function is proposed to dynamically change grid points. Through the simulation experiments against various disturbances, all of our final guidance errors are less than 1 km, and waste times are less than 135.62s, which can proved the effectiveness and robustness of our method. Compared with the conventional Gaussian Spectral method, it can obtain the comparable accuracy and faster generation speed.
And the revised parts have been marked in yellow.

Reviewer 2 Report
An trajectory generation algorithm based on convex programming for reentry glide vehicle is proposed in this paper. And there are three innovations proposed. Firstly, all the dynamic model is transformed into the flight range domain to realize convexation and discretization; Secondly, an initial trajectory generation method is proposed which can generate the initial trajectory better; Thirdly, an distributed grid points adjustment method is proposed to adjust mesh points rapidly. On the whole, the idea of this paper is pretty nice and have much engineering application value, and the formula derivation is detailed. And there are several minor problems that need to be answered.
1. Please add the description about the reason why the flight range domain described in this paper is more suitable for trajectory generation of reentry glide vehicle.
2. The grid points generated is generated in this paper do not change slightly, please give a reasonable explanation
3. Please discuss the potential improvements and future work in the appropriate sections, such as the cooperative guidance issues
Cooperative guidance strategy for multiple hypersonic gliding vehicles system, Chinese Journal of Aeronautics.
Adaptive Practical Optimal Time-Varying Formation Tracking Control for Disturbed High-Order Multi-Agent Systems.
4. There are some minor typos. Please revise and polish the language of the paper carefully
Author Response
Dear reviewer:
Thank you for your letter and the comments concerning our manuscript entitled “A trajectory generation algorithm for reentry gliding vehicle based on convex optimization in the flight range domain and distributed grid points adjustment” (ID: appisci 2169933). Those comments are all valuable and helpful for revising and improving our papers, as well as the important guiding significance to our researches. We have studied comments carefully and have made corrections which we hope meeting with approval. The main corrections in the paper and the responds to the comments are as follows:
Q1: Please add the description about the reason why the flight range domain described in this paper is more suitable for trajectory generation of reentry glide vehicle.
Reply: Because the trajectory generation problem of the reentry glide vehicle can be regarded as a guidance problem, when the vehicle gradually approaches the target point, its projected range-to-go gradually approaches 0. And the projected range-to-go is a constants for different initial and termination points, which is easy to determine the position of grid points. However, in the time domain, the total time of the trajectory generation problem of the vehicle is variable, and the time needs to be de scaled.
Q2: The grid points generated is generated in this paper do not change slightly, please give a reasonable explanation
Reply: In order to improve the running speed of the algorithm, the total number of grid points is limited in the paper, which makes the nonlinear illegal degrees of most of the initial grid points res high, and the number of grid points that can be adjusted is small, resulting in slight changes of grid points.
Q3: Please discuss the potential improvements and future work in the appropriate sections, such as the cooperative guidance issues
Cooperative guidance strategy for multiple hypersonic gliding vehicles system, Chinese Journal of Aeronautics.
Adaptive Practical Optimal Time-Varying Formation Tracking Control for Disturbed High-Order Multi-Agent Systems.
Reply: Related potential improvements and future work has been added in our manuscript, which is marked in blue.
Q4: There are some minor typos. Please revise and polish the language of the paper carefully
Reply: The whole manuscript has been revised and poliseded.

Reviewer 3 Report
Interesting research about trying to increase the solution rate of numerical optimum motion trajectory generation, and the work is assembled into a decently drafted manuscript that needs some mild revisions.
· The manuscript is clear, relevant for the field and presented in a well-structured manner. The cited references are current (some few within the last 5 years), while suggested very recently published references are suggested in review to aid revision. The manuscript is scientifically sound, and the experimental design is appropriate to test the hypothesis. The manuscript’s results are reproducible based on the details given in the methods section. The figures/tables/images/schemes appropriate and properly show the data. They are easy to interpret and understand. The data is interpreted appropriately and consistently throughout the manuscript. The conclusions are consistent with the evidence and arguments presented.
The Abstract is okay but is not likely to entice the readership to continue reading the rest of the manuscript.
· Results are only presented in a weak, qualitative fashion. Highest quality expression of main conclusions or interpretations is quantitative results discussed in the broadest context possible, e.g., percent performance improvement compared to a declared benchmark. “…prove the effectiveness and robustness of our method.…” is very weakly stated results compared to “…xxx percent performance improvement over conventional methods was achieved….”
The Introduction is decently done with some omitted very recent literature and some very mild, acceptable abuse of multi-citation without elaboration (only 2 double-citations without elaboration).
· Just last march in an already highly cited work, Sandberg proposed several autonomous trajectory generation algorithms, and some of those proposed included convex optimization, while the manuscript provides direct comparison of the disparate approaches in https://doi.org/10.3390/aerospace9030135.
· Shortly afterwards, Raigoza augmented Sandberg’s approach with distributed waypoint for autonomous collision avoidance in https://doi.org/10.3390/s22187066.
Equations are scientifically sound and well presented, enhancing the manuscript quality.
Figures are decently done with some mandatory improvements to ensure the readership has access to the content.
· Internal font size is generally well done. The legends of figures are nearly illegibly small.
· Line styles and sizes are identical in figures 1,5, 9 rendering the disparate data indistinguishable when the manuscript is read in printed hardcopy (particularly in black and white) negating the value of the figures due to reliance on colors.
Tables are decently done to introduce problem formation (aiding repeatability), but quantitative results are neglected.
· Particularly for comparative figures (e.g., 12), please add a table of accompanying canonical figures of merit (e.g., means and deviations of difference, or others) to help the reader ascertain quantitative differences between the plotted data.
· For such a manuscript, heavy in acronym and variable usage, thanks for adding the table of definitions in the appendix. Periodic mini tables of proximal acronyms and variables would increase readability.
Author Response
Dear reviewer and editor:
Thank you for your letter and the comments concerning our manuscript entitled “A trajectory generation algorithm for reentry gliding vehicle based on convex optimization in the flight range domain and distributed grid points adjustment” (ID: appisci 2169933). Those comments are all valuable and helpful for revising and improving our papers, as well as the important guiding significance to our researches. We have studied comments carefully and have made corrections which we hope meeting with approval. The main corrections in the paper and the responds to the comments are as follows:
Q1: The Abstract is okay but is not likely to entice the readership to continue reading the rest of the manuscript.
Reply: The abstract has been rewritten to make the manuscript more readable, and the revised parts have been marked in green.
Q2: Results are only presented in a weak, qualitative fashion. Highest quality expression of main conclusions or interpretations is quantitative results discussed in the broadest context possible, e.g., percent performance improvement compared to a declared benchmark. “…prove the effectiveness and robustness of our method.…” is very weakly stated results compared to “…xxx percent performance improvement over conventional methods was achieved….”.
Reply: We have revised the description of results with the higher quality expression, and revised parts have been marked in green.
Q3: Just last march in an already highly cited work, Sandberg proposed several autonomous trajectory generation algorithms, and some of those proposed included convex optimization, while the manuscript provides direct comparison of the disparate approaches in https://doi.org/10.3390/aerospace9030135. Shortly afterwards, Raigoza augmented Sandberg’s approach with distributed waypoint for autonomous collision avoidance in https://doi.org/10.3390/s22187066.
Reply: Several relative literatures have been considered and cited in our manuscript. The revised parts have been marked in green.
Q4: Internal font size is generally well done. The legends of figures are nearly illegibly small.
Reply: The legends of figures have been revised. The revised parts have been marked in green.
Q5: Line styles and sizes are identical in figures 1,5,9 rendering the disparate data indistinguishable when the manuscript is read in printed hardcopy (particularly in black and white) negating the value of the figures due to reliance on colors.
Reply: The figure 1,5,9 have been generated again with the different line and mark type, which can be distinguishable. The revised parts have been marked in green.
Q6: Tables are decently done to introduce problem formation (aiding repeatability), but quantitative results are neglected.
Reply: The description and explanation of quantitative results have been added in our manuscript. The revised parts have been marked in green;
Q7: Particularly for comparative figures (e.g., 12), please add a table of accompanying canonical figures of merit (e.g., means and deviations of difference, or others) to help the reader ascertain quantitative differences between the plotted data
Reply: We have added a table to compare the difference between our method and other method. The revised parts have been marked in green.
Q8: For such a manuscript, heavy in acronym and variable usage, thanks for adding the table of definitions in the appendix. Periodic mini tables of proximal acronyms and variables would increase readability.
Reply: The definition table of the acronym and variables has been added in appendix. The revised parts have been marked in green.
